# Efficient Whole-Body Tumor Segmentation with a 5.6M Parameter 3D U-Net

Ziyan Huang[1][0000−0002−1533−5239]

Shanghai Jiao Tong University {ziyanhuang}@sjtu.edu.cn

**Abstract.** Accurate organ and cancer segmentation in medical imaging, especially in 3D CT scans, is essential for precise diagnosis, treatment planning, and disease monitoring. The MICCAI FLARE24 challenge aims to advance pan-cancer segmentation algorithms, with Task 1 focusing on whole-body cancer segmentation in CT scans. In this paper, we present an efficient approach utilizing a lightweight 3D U-Net architecture to address this challenge. Our model comprises only four resolution stages and approximately 5.6 million parameters, significantly reducing computational demands while maintaining performance. Our method processed the 279 CT images in the public validation set in just 18 minutes, averaging under 4 seconds per scan. For individual cases, the average prediction time per scan was approximately 20 seconds. We achieved an average Dice Similarity Coefficient (DSC) of 25.34% and a Normalized Surface Dice (NSD) of 24.40% on the public validation set.

## 1 Introduction

Whole-body tumor segmentation from CT scans is a critical yet challenging task in medical image analysis. The MICCAI FLARE24 challenge, specifically Task 1, aims to advance the field of pan-cancer segmentation by addressing this complex problem. The difficulty of this challenge arises from several factors:

– **Anatomical Diversity**: The vast anatomical diversity across different patients and the varied appearance of tumors in different body regions make it challenging to develop a universally effective segmentation model.
– **Computational Demands**: Whole-body CT scans are large, often containing thousands of slices. Processing these scans in real-time or near-real-time settings poses significant computational challenges.
– **Partial Labeling**: The dataset's partial labeling nature, where not all tumors are annotated in every scan, adds another layer of complexity to the learning process.

Recent advancements in deep learning have led to significant progress in medical image segmentation. In the context of semi-supervised and partial-label segmentation, several approaches have shown promise:

– **Self-training Methods**: These methods iteratively use the model's own predictions on unlabeled data to augment the training set, gradually improving performance [11].

– **Consistency-based Approaches**: These leverage the principle that model predictions should remain consistent under different perturbations of the input or model [25].
– **Adversarial Learning**: Some methods employ adversarial training to align the distributions of labeled and unlabeled data, enhancing generalization [23].
– **Multi-task Learning**: By combining segmentation with auxiliary tasks like reconstruction or classification, these methods can better utilize partially labeled data [3].

While these approaches have achieved success in various medical imaging tasks, they often require significant computational resources, limiting their applicability in real-world clinical scenarios.

Motivated by the need for an efficient yet accurate solution to whole-body tumor segmentation, we propose a lightweight 3D U-Net architecture. Our approach is guided by the following key considerations:

– **Efficiency**: In clinical settings, fast processing times are crucial. We aim to develop a model that can segment whole-body CT scans in near real-time while maintaining high accuracy.
– **Resource Constraints**: Many healthcare facilities may lack access to high-end GPU resources. Our lightweight model addresses this by minimizing computational requirements.
– **Generalizability**: Given the diverse nature of whole-body tumors, we focus on creating a model that can generalize well across different anatomical regions and tumor types.

Our main contributions are:

(1) A compact 3D U-Net architecture with only 5.6 million parameters, significantly reducing the model size compared to standard implementations.
(2) An efficient segmentation approach that balances accuracy and speed, processing whole-body CT scans in approximately 10 seconds.
(3) Comprehensive evaluation on a large-scale dataset of over 10,000 CT scans, demonstrating the model's effectiveness across diverse tumor types and anatomical locations.
(4) Insights into the trade-offs between model complexity and performance in the context of whole-body tumor segmentation, potentially guiding future research in this area.

By addressing the challenges of whole-body tumor segmentation with a focus on efficiency and practicality, our work aims to bridge the gap between advanced AI models and real-world clinical applications.

## 2   Method

### 2.1   Preprocessing

We utilized the default preprocessing pipeline of `nnU-Net` [12]. The preprocessing steps included:

– **Dataset Selection**: Only images with tumor annotations were used for training.
– **Resampling**: All CT scans were resampled to an isotropic voxel spacing of $2 \times 2 \times 3$ mm$^3$.
– **Intensity Normalization**: Hounsfield Unit (HU) values were clipped at the 0.05% and 99.5% percentiles and then normalized by subtracting the mean and dividing by the standard deviation, based on statistics computed from the entire dataset.

## 2.2   Proposed Method

Figure 1 illustrates the architecture of our proposed model in comparison to deeper variants. Unlike conventional U-Net or nnU-Net structures, which typically employ 5 or even 6 resolution stages, our model adopts a shallower 4-stage design. As the later stages in U-Net architectures generally involve high channel dimensions and incur substantial parameter costs, our streamlined structure significantly reduces the total number of parameters without compromising segmentation performance. Apart from the number of stages, the rest of our model architecture remains aligned with the standard nnU-Net framework.

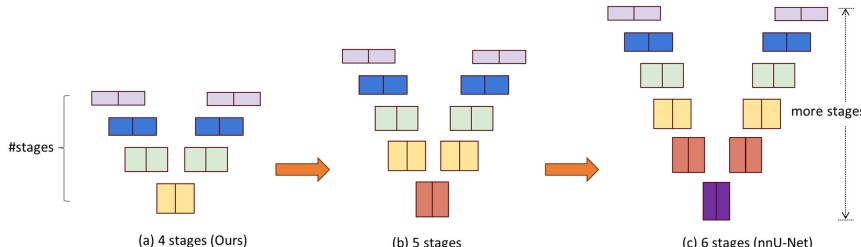

**Fig. 1.** Architectures with different stage numbers: (a) our 4-stage model, (b) a 5-stage variant, and (c) the 6-stage nnU-Net. More stages lead to deeper models and higher complexity.

Loss function: we use the summation between Dice loss and cross-entropy loss because compound loss functions have proven robust in various medical image segmentation tasks [14].

We did not employ specific strategies to reduce false positives on CT scans from healthy patients, handle partial labels, or use unlabeled images. We only used the provided labeled data without distinguishing whether it was partially labeled, and unlabeled images were not used. Additionally, we did not use the pseudo labels generated by the FLARE23 winning algorithm.

We used a smaller model architecture and resampled the images to a relatively low resolution of $2\times2\times3$ mm$^3$, which helped to improve inference speed and reduce resource consumption during the inference phase.

## 3     Experiments

### 3.1     Dataset and evaluation measures

The segmentation targets cover various lesions. The training dataset is curated from more than 50 medical centers under the license permission, including TCIA [4], LiTS [2], MSD [22], KiTS [8,10,9], autoPET [7,6], TotalSegmentator [24], and AbdomenCT-1K [19], FLARE 2023 [18], DeepLesion [27], COVID-19-CT-Seg-Benchmark [17], COVID-19-20 [21], CHOS [13], LNDB [20], and LIDC [1]. The training set includes 4000 abdomen CT scans where 2200 CT scans with partial labels and 1800 CT scans without labels. The validation and testing sets include 100 and 400 CT scans, respectively, which cover various abdominal cancer types, such as liver cancer, kidney cancer, pancreas cancer, colon cancer, gastric cancer, and so on. The lesion annotation process used ITK-SNAP [28], nnU-Net [12], MedSAM [15], and Slicer Plugins [5,16].

The evaluation metrics encompass two accuracy measures—Dice Similarity Coefficient (DSC) and Normalized Surface Dice (NSD)—alongside two efficiency measures—running time and area under the GPU memory-time curve. These metrics collectively contribute to the ranking computation. Furthermore, the running time and GPU memory consumption are considered within tolerances of 45 seconds and 4 GB, respectively.

### 3.2     Implementation details

**Environment settings** The development environments and requirements are presented in Table 1.

**Table 1.** Development environments and requirements.

| | |
|---|---|
| System version | CentOS Linux release 7.6.1810 |
| CPU | Dual AMD Rome 7742@3.4GHz |
| RAM | 32×32GB; 3200MT/s |
| GPU (number and type) | 1x NVIDIA A100 80GB Tensor Core GPUs |
| CUDA version | 11.2 |
| Programming language | Python 3.8.0 |
| Deep learning framework | Pytorch (Torch 1.10.1) |
| Specific dependencies | nnU-Net 2.2.0 |
| Code | https://github.com/Ziyan-Huang/FLARE24 |

**Training Protocols** Our training protocols followed the default settings of nnU-Net.

1. Specifically, no specific strategies were applied for the processing of unlabeled images and partial labels; we used only the provided labeled data without distinguishing between partially labeled or unlabeled data.

2. For data augmentation, we employed the default extensive augmentation techniques as implemented in nnU-Net.

3. The patch sampling strategy was also based on the default configuration of nnU-Net.

4. Additionally, we did not perform specific model selection, and training continued until the final epoch without early stopping or optimal model selection.

**Table 2.** Training protocols.

| | |
|---|---|
| Network initialization | He |
| Batch size | 2 |
| Patch size | 128×128×96 |
| Total epochs | 2000 |
| Optimizer | SGD |
| Initial learning rate (lr) | 0.001 |
| Lr decay schedule | poly |
| Training time | 20 hours |
| Loss function | Dice plus CE |
| Number of model parameters | 5.6M[1] |
| Number of flops | 59.32G[2] |
| $CO_2$eq | 1 Kg[3] |

# 4   Results and discussion

**Table 3.** Quantitative evaluation results.

| Methods | Public Validation | | Online Validation | | Testing | |
|---|---|---|---|---|---|---|
| | DSC(%) | NSD(%) | DSC(%) | NSD(%) | DSC(%) | NSD (%) |
| Ours | 25.34 ± 31.56 | 24.40 ± 27.80 | – | – | 43.11 ± 39.98 | 35.67 ± 35.76 |

## 4.1   Quantitative results on validation set

Quantitative results are shown in Table 3. On the public validation set, our method achieved a mean Dice of 25.34% and NSD of 24.40%. On the hidden test set, performance improves to 43.11% Dice and 35.67% NSD, showing good generalization. Among 40 healthy cases, 16 showed false positives, mostly in ambiguous soft tissue regions.

### 4.2   Qualitative results on validation set

Figure 2 shows two successful and two failed tumor segmentation cases. In the good cases (top two rows), the model accurately captures the tumor region, achieving high Dice scores (e.g., 0.9451). In contrast, the failed cases (bottom two rows) show missed detections or incomplete coverage of tumor areas, with Dice scores below 0.3. These failures are often due to small tumor size, low contrast, or surrounding complex textures.

### 4.3   Segmentation efficiency results on validation set

Table 4 shows inference efficiency on representative cases. Average runtime is around 22 seconds per scan, with peak GPU usage below 5.2 GB, showing good scalability.

**Table 4.** Quantitative evaluation of segmentation efficiency in terms of the running them and GPU memory consumption. Total GPU denotes the area under GPU Memory-Time curve.

| Case ID | Image Size | Running Time (s) | Max GPU (MB) | Total GPU (MB) |
|---------|------------|------------------|--------------|----------------|
| 0001 | (512, 512, 55) | 28.51 | 4696 | 141667 |
| 0051 | (512, 512, 100) | 18.85 | 4916 | 92644 |
| 0017 | (512, 512, 150) | 18.87 | 5170 | 97570 |
| 0019 | (512, 512, 215) | 19.47 | 4930 | 95986 |
| 0099 | (512, 512, 334) | 21.82 | 5177 | 112956 |
| 0063 | (512, 512, 448) | 18.16 | 4776 | 86711 |
| 0048 | (512, 512, 499) | 25.17 | 5017 | 126278 |
| 0029 | (512, 512, 554) | 24.57 | 4639 | 113978 |

### 4.4   Results on final testing set

Table 5 summarizes the final testing results of our method submitted to the FLARE23 challenge. The metrics include Dice Similarity Coefficient (DSC), Normalized Surface Dice (NSD), inference time, and GPU memory usage. Our method achieved competitive segmentation accuracy while maintaining relatively efficient runtime and resource consumption.

### 4.5   Limitation and Future Work

Despite achieving efficiency, our model has several limitations:

(1) **Accuracy**: The DSC and NSD scores suggest there is room to improve segmentation accuracy, particularly for small or irregular tumors.

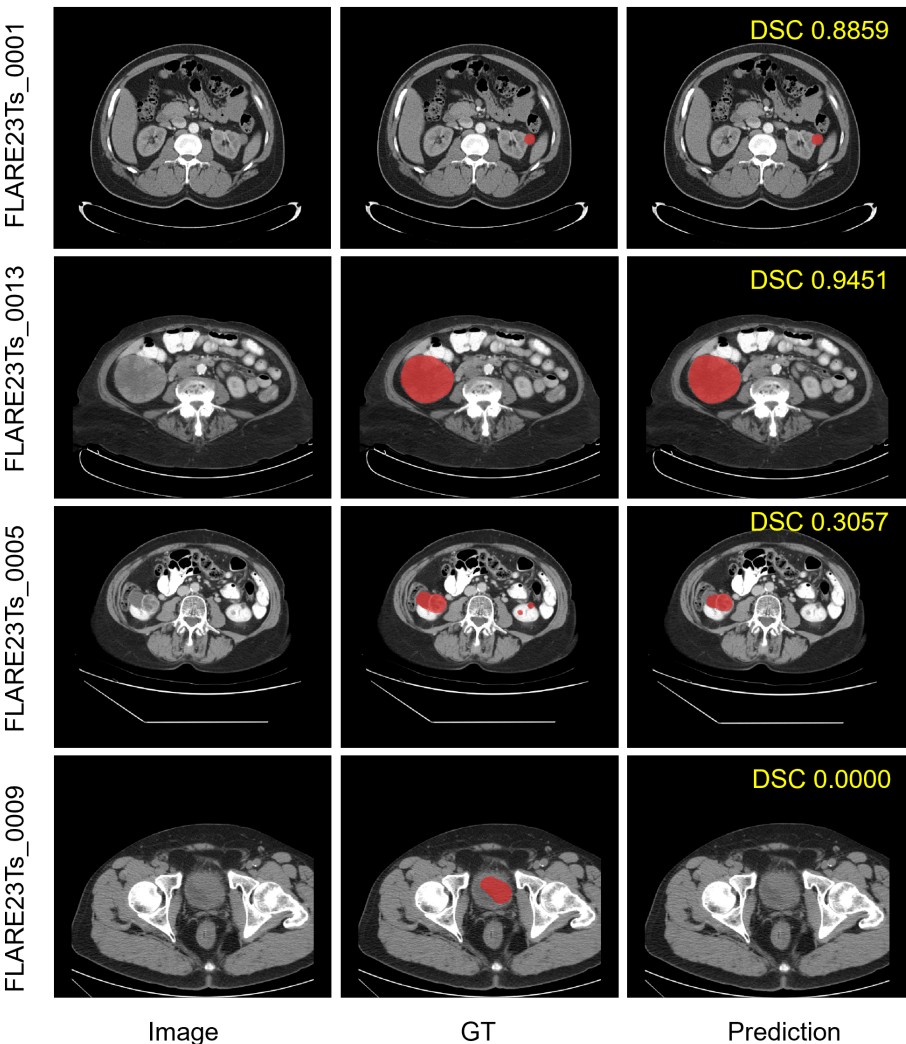

**Fig. 2.** Two examples with good segmentation results and two examples with bad segmentation results in the validation set.

**Table 5.** Final testing results on the FLARE23 challenge (team: gmai).

| | |
|---|---|
| **Avg DSC** | 43.11 ± 39.98 |
| **Median DSC** | 47.11 (0.00, 85.15) |
| **Avg NSD** | 35.67 ± 35.76 |
| **Median NSD** | 30.68 (0.00, 70.06) |
| **Avg Time (s)** | 32.46 ± 14.32 |
| **Median Time (s)** | 28.39 (20.62, 40.23) |
| **Avg GPU (MB)** | 43451.6 ± 19941.9 |
| **Median GPU (MB)** | 39401.5 (28946.0, 57435.5) |

(2) **Partial Labels**: We did not utilize unlabeled or partially labeled data, limiting generalization. Future work will incorporate semi-supervised learning or pseudo-labeling.

(3) **False Positives**: High false positive rates in healthy cases reduce clinical reliability. Advanced post-processing methods will be explored.

## 5   Conclusion

We presented a 5.6M parameter 3D U-Net model for efficient whole-body tumor segmentation. The model achieved a DSC of 25.34% and NSD of 24.40%, with an inference time of less than 4 seconds per scan, highlighting its suitability for near real-time applications. Key contributions include reducing model complexity while maintaining a balance between speed and accuracy. Future work will focus on improving segmentation precision and expanding the model's applicability across diverse clinical settings.

**Acknowledgements** The authors of this paper declare that the segmentation method they implemented for participation in the FLARE 2024 challenge has not used any pre-trained models nor additional datasets other than those provided by the organizers. The proposed solution is fully automatic without any manual intervention. We thank all data owners for making the CT scans publicly available and CodaLab [26] for hosting the challenge platform.

## Disclosure of Interests

The authors declare no competing interests.

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
