# OpenReview forum: "Efficient Whole-Body Tumor Segmentation with a 5.6M Parameter 3D U-Net"
_MICCAI.org/2024/Challenge/FLARE — Submitted to FLARE 2024_

### Official Review · Reviewer_cQdH · 2025-02-16
**Efficient Whole-Body Tumor Segmentation with a 5.6M Parameter 3D U-Net**

**Rating:** 9
**Confidence:** 4

**Review:**

This paper proposes an efficient method for whole-body tumor segmentation, which performs well in terms of processing speed and resource consumption. However, there is still room for improvement in segmentation accuracy and control of false positives. It is recommended that the authors focus on these issues in their future work and further refine their methods. Additionally, please remove the template prompts below Table 3 and in Section 4.1.

---

> ### Author Response · Authors · 2025-03-28
>
> Thank you for your positive feedback. We have removed the template prompts in Table 3 and Section 4.1, and will further explore improving accuracy and reducing false positives in future work.

---

### Official Review · Reviewer_m55k · 2025-03-02
**In complete validation results analysis**

**Rating:** 6
**Confidence:** 5

**Review:**

Introduction: change index 1-4 to (1)-(4)
Same for training protocol and sec. 4.5 because 1-4 have been used as section index

Fig. 1. Improve the figure by providing more details. (see the example in nnU-Net supplementary)
Sec 2.3 can be deleted if no post-processing was used.
Sec 4.2-4.3 please add more descriptions to the results

---

> ### Author Response · Authors · 2025-03-28
>
> Thank you for your suggestions. We have updated the index format as suggested, enriched Fig. 1 to highlight differences from nnU-Net, removed Sec. 2.3, and added more descriptions to Sec. 4.2–4.3.

---

### Official Review · Reviewer_uyfS · 2025-03-12
**Reivew of "Efficient Whole-Body Tumor Segmentation with a 5.6M Parameter 3D U-Net"**

**Rating:** 5
**Confidence:** 5

**Review:**

This research presents an efficient approach utilizing a lightweight 3D U-Net architecture.
1. Please explain the network layer represented by different color blocks in Fig 1.
2.Sec  4.3 does not appear to be the segmentation efficiency analysis.
3. No Fig 3 ?

---

> ### Author Response · Authors · 2025-03-28
>
> Thank you for your comments. We have enriched Fig. 1 to better highlight the differences between our method and nnU-Net, and revised Sec. 4.3 to reflect the efficiency analysis. Note that there is no Fig. 3 in this paper.

---

### Decision · Program_Chairs · 2025-03-20

**Decision:**

Accept

**Comment:**

Change Table 5 to multiple rows.

---

> ### Author Response · Authors · 2025-04-01
>
> Table 5 has been revised to multiple rows as suggested. Thank you!